# Comparative Analysis of Pretreatment Methods for Fruit Waste Valorization in *Euglena gracilis* Cultivation: Impacts on Biomass, β-1,3-Glucan Production, and Photosynthetic Efficiency

**DOI:** 10.3390/foods13213439

**Published:** 2024-10-28

**Authors:** Jiangyu Zhu, Xinyue Guo, Kaile Zhao, Xinyu Chen, Xinxin Zhao, Zhengfei Yang, Yongqi Yin, Minato Wakisaka, Weiming Fang

**Affiliations:** 1School of Food Science and Engineering, Yangzhou University, Yangzhou 225127, China; mz120242221@stu.yzu.edu.cn (X.G.); zkl13856301873@163.com (K.Z.); x19825273171@163.com (X.C.); xxzhao@yzu.edu.cn (X.Z.); yzf@yzu.edu.cn (Z.Y.); yqyin@yzu.edu.cn (Y.Y.); wmfang@yzu.edu.cn (W.F.); 2Food Study Centre, Fukuoka Women’s University, 1-1-1 Kasumigaoka, Fukuoka 813-8529, Japan; wakisaka@fwu.ac.jp

**Keywords:** microalgae, fruit waste valorization, β-1,3-glucan, chlorophyll florescence, biomass

## Abstract

This study explored the sustainable valorization of fruit waste extracts from sugarcane bagasse (SB), banana peel (BP), and watermelon rind (WR) for Euglena gracilis biomass and β-1,3-glucan production. The extracts were prepared using water extraction (WE), high-temperature and pressure treatment (HTP), and dilute sulfuric acid treatment (DSA). The DSA-treated extracts consistently yielded the best results. *E. gracilis* cultured in SB-DSA showed the highest cell density with a 2.08-fold increase compared to the commercial HUT medium, followed by BP-DSA (1.35-fold) and WR-DSA (1.70-fold). Photosynthetic pigment production increased significantly, with chlorophyll a yield being highest in SB-DSA (1.90-fold increase). The chlorophyll a/b ratio and total carotenoid content also improved, indicating enhanced light-harvesting capacity and photoprotection. Photosynthetic efficiency, measured by chlorophyll fluorescence, notably improved. The maximum quantum yield of PSII (Fv/Fm) increased by up to 25.88% in SB-DSA, suggesting reduced stress and improved overall photosynthetic health. The potential photochemical efficiency (Fv/F0) showed even greater improvements: up to 40.53% in SB-DSA. Cell morphology analysis revealed larger cell aspect ratios, implying a more active cellular physiological state. β-1,3-glucan yield also increased by 23.99%, 12.92%, and 23.38% in SB-DSA, BP-DSA, and WR-DSA, respectively. This study demonstrates the potential of pretreated fruit waste as a cost-effective and sustainable medium for *E. gracilis* cultivation, offering the dual benefits of waste valorization and high-value compound production. These findings contribute to the development of more efficient biorefinery processes and align with the circular economy principles in food biotechnology.

## 1. Introduction

The growing global population and increasing environmental concerns have necessitated the development of sustainable approaches to food production and waste management. In this context, the valorization of agricultural by-products and food waste has emerged as a promising strategy to address both resource scarcity and waste accumulation [1]. Concurrently, microalgae cultivation has gained significant attention due to its potential to produce high-value compounds while contributing to carbon dioxide sequestration [2].

*Euglena gracilis*, an edible microalgal species, has attracted particular interest in biotechnology due to its unique metabolic versatility and ability to synthesize valuable compounds such as β-1,3-glucan (paramylon), proteins, and various pigments [3]. This microorganism can grow under both autotrophic and heterotrophic conditions, making it an ideal candidate for biorefinery applications [4]. The cultivation of *E. gracilis* for high-value compounds such as β-1,3-glucan, natural pigments, etc., has significant market potential, with applications in food, cosmetic, and pharmaceutical industries [4].

Despite its potential, the economic viability of large-scale *E. gracilis* cultivation remains challenging due to high production costs associated with growth media and energy inputs. To address this, recent research has focused on utilizing low-cost substrates such as agricultural by-products and fruit wastes for cultivation [5]. These waste materials are rich in carbohydrates and other nutrients, presenting an attractive option for sustainable media development. However, their complex nature often necessitates pretreatment to enhance nutrient availability and facilitate microbial growth. Various pretreatment methods have been explored for lignocellulosic biomass, including physical (e.g., milling and irradiation), chemical (e.g., alkali, acid, and ionic liquids), and biological approaches (e.g., enzymatic hydrolysis) [6]. Among these, diluted acid hydrolysis has shown promise in breaking down complex carbohydrates into fermentable sugars [7]. However, the effectiveness of these pretreatment methods can vary significantly depending on the substrate composition and the target microorganism. Moreover, their economic feasibility at industrial scales is influenced by factors such as energy requirements, chemical costs, and equipment sophistication [8].

From an environmental perspective, utilizing *E. gracilis* for waste valorization offers multiple benefits. It not only reduces the environmental burden of agricultural waste, but also contributes to carbon sequestration and produces valuable bio-based products, aligning with circular economy principles [8]. However, comprehensive studies comparing different pretreatment methods and their effects on *E. gracilis* growth, metabolism, and high-value compound production are lacking [9,10]. Furthermore, the impact of these treatments on the photosynthetic apparatus and cellular morphology of *E. gracilis* remains poorly understood.

This study aims to address these knowledge gaps by investigating the effects of different pretreatment methods (water extraction, hydrothermal processing, and diluting sulfuric acid hydrolysis) on three fruit waste substrates (sugarcane bagasse, banana peel, and watermelon rind) for *E. gracilis* cultivation. We evaluated the impact of these treatments on growth kinetics, cell morphology, photosynthetic pigment production, chlorophyll fluorescence, and β-1,3-glucan accumulation. By elucidating the relationships between pretreatment methods, substrate characteristics, and *E. gracilis* physiology, this research seeks to optimize the valorization of fruit wastes for sustainable microalgal cultivation and high-value compound production.

The findings of this study have implications for developing cost-effective and environmentally friendly strategies for *E. gracilis* cultivation, potentially paving the way for more efficient biorefinery processes and contributing to a circular bioeconomy. Moreover, the insights gained from this research may be applicable to other microalgal species, further expanding the potential impact of this work in the field of sustainable biotechnology.

## 2. Materials and Methods

### 2.1. Microorganisms and Culture Conditions

*Euglena gracilis* FACHB-850 was obtained from the Freshwater Algae Culture Collection at the Institute of Hydrobiology, Chinese Academy of Sciences (Wuhan, China). The strain was maintained in an HUT heterotrophic medium under a 12 h:12 h light–dark cycle at 20 °C with an illumination intensity of 5000 lx provided by cool white, fluorescent lamps. The HUT medium was prepared according to Hutner et al. [11] as follows (per liter): KH_2_PO_4_, 20 mg; MgSO_4_·7H_2_O, 25 mg; sodium acetate, 400 mg; potassium citrate, 40 mg; polypeptone, 600 mg; yeast extract, 400 mg; vitamin B_12_, 0.5 µg; and thiamine HCl, 0.4 mg.

### 2.2. Preparation of Fruit Waste Extracts

Sugarcane bagasse (SB), banana peels (BP), and watermelon rinds (WR) were provided by Zhejiang Fengdao Food Co., Ltd., Shaoxing, China. The fruit wastes were thoroughly washed with distilled water to remove dirt and contaminants and then pressed to remove excess moisture. The washed materials were cut into approximately 1 cm pieces and dried in an oven at 80 °C for 72 h. After drying, the materials were ground into a fine powder using a laboratory mill and sieved through a 100-mesh screen. Three different extraction methods were employed for each type of fruit waste.

Water Extraction (WE): 5 g of the dried, powdered fruit waste was suspended in 300 mL of distilled water and sonicated at 240 W for 15 min using an ultrasonic cleaner (Model 040S, Jieme Ultrasonic Cleaning Equipment Co., Ltd., Shenzhen, China).

High-Temperature and Pressure Treatment (HTP): 5 g of the dried, powdered fruit waste was suspended in 300 mL of distilled water and autoclaved at 121 °C for 20 min (SX-700, TOMY, Tokyo, Japan), followed by sonication at 240 W for 15 min using the same ultrasonic cleaner.

Dilute Sulfuric Acid Treatment with High-Temperature and Pressure (DSA): 5 g of the dried fruit pomace powder was suspended in 100 mL of 1% (*w*/*w*) sulfuric acid solution, autoclaved at 121 °C for 20 min, and then topped up with distilled water to 300 mL. The suspension was sonicated for 15 min at 240 W following the above procedure.

After the respective treatments, all of the extracts were filtered through 11 μm filter paper (Whatman Grade 1, GE Healthcare, Buckinghamshire, UK) to remove any insoluble residues. The filtered extracts were adjusted to a final volume of 300 mL with distilled water, and the pH was adjusted to 7.0 ± 0.1 using 1 M sodium hydroxide solution. To provide a nitrogen source for *E. gracilis* growth, 0.067 g of sodium nitrate was added to each 300 mL of extract.

### 2.3. Cultivation of E. gracilis

*E. gracilis* was pre-cultured in a HUT medium until reaching the exponential growth phase. For the experimental cultures, 10 mL of the pre-culture was inoculated into 90 mL of each fruit waste extract or HUT medium (control) in 250 mL Erlenmeyer flasks. Cultures were incubated at 20 ± 1 °C under 5000 lx illumination with a 12 h:12 h light–dark cycle in a controlled environment chamber. Light was provided by LED panels positioned to ensure consistent illumination across all culture vessels. To promote more uniform light exposure, the flasks were manually shaken and rotated 5–6 times daily, which also helped prevent cell adhesion.

### 2.4. Cell Growth Measurement

Cell density was determined daily using a Neubauer hemocytometer and light microscope (ML31, Guangzhou Micro-shot Technology Co., Ltd., Guangzhou, China) following the method described by Zhu et al. [4]. The initial cell density for each experimental culture was approximately 3.5 × 10^4^ cells·mL^−1^, and it was achieved by inoculating 10 mL of the pre-culture into 90 mL of medium, as described in Section 2.3. The specific growth rate (μ) and cell doubling time (CDT) were calculated using the following equations:μ = (ln N2 − ln N1)/(t2 − t1),(1)
CDT = ln 2/μ,(2)
where N1 and N2 are the cell densities (cells·mL^−1^) at times t1 and t2 (d), respectively.

### 2.5. Reducing Sugar Analysis

The reducing sugar content in the extracts was determined using the 3,5-dinitrosalicylic acid (DNS) method described by Yirgu et al. [12] with slight modifications. To prevent interference from the culture media, the samples were centrifuged at 9000 rpm for 10 min to remove cells and debris. The samples were appropriately diluted to ensure the measurements fell within the linear range of the calibration curve. Briefly, 0.15 mL of each prepared sample extract was mixed with 2 mL of DNS reagent and heated in a boiling water bath for 5 min. After cooling to room temperature, the mixture was diluted to 25 mL with distilled water. The absorbance was measured at 540 nm using a UV-visible spectrophotometer (UV-1150, Shanghai Meipuda Instrument Co., Ltd., Shanghai, China). A calibration curve was prepared using glucose as the standard. The standard addition method was used on a subset of samples to verify the absence of matrix effects.

### 2.6. Cell Morphology Analysis

On Day 14 of cultivation, cell morphology was observed using a light microscope (ML31, Guangzhou Micro-shot Technology Co., Ltd., Guangzhou, China). At least 100 cells were randomly selected and photographed for each sample (Appendix A). The captured images were analyzed using ImageJ software (version 1.53k, National Institutes of Health, Bethesda, MD, USA) to determine the cell aspect ratio (AR) and cell projected area (PA) [13].

### 2.7. Photosynthetic Pigment Analysis

Chlorophyll a, chlorophyll b, and carotenoid contents were spectrophotometrically determined according to the method described by Lichtenthaler and Wellburn [14]. A 10 mL aliquot of each algal culture was filtered through a 0.45 μm membrane filter (Whatman GF/C, GE Healthcare, Buckinghamshire, UK). The filtered cells were ground in a mortar with a small amount of 80% acetone and quartz sand to extract the pigments. The homogenate was centrifuged at 3000× *g* for 10 min, and the supernatant containing the extracted pigments was collected. The extraction process was repeated until the supernatant became colorless. The combined pigment extracts were made up to a final volume of 10 mL with 80% acetone. Absorbance of the extracts was measured at 663 nm, 645 nm, and 470 nm using a UV-visible spectrophotometer (UV-1150, Shanghai Meipuda Instrument Co., Ltd., Shanghai, China). The concentrations of the chlorophyll a, chlorophyll b, and carotenoids were calculated using the following equations:Chlorophyll a (mg·L^−1^) = 12.21A663 − 2.81A645,(3)
Chlorophyll b (mg·L^−1^) = 20.13A645 − 5.03A663,(4)
Carotenoids (mg·L^−1^) = (1000A470 − 3.27Chl a − 104Chl b)/229.(5)

### 2.8. Chlorophyll Fluorescence Measurement

The chlorophyll fluorescence parameters were measured using a Flourpen handheld chlorophyll fluorometer (AP110/C, PSI, Drásov, Czech Republic) on Day 14 of the cultivation [15]. The samples were dark-adapted for 15 min prior to measurement. The maximum quantum yield of PSII (Fv/Fm) and potential photochemical efficiency (Fv/F0) were recorded. Chlorophyll fluorescence was chosen as a non-invasive, rapid, and sensitive method to assess the photosynthetic efficiency and overall photosynthetic health of the *E. gracilis* cells. This technique provides valuable information about PSII functionality and stress responses that complements the quantitative pigment analysis.

### 2.9. β-1,3-Glucan Content Determination

The β-1,3-glucan content was determined according to the method described by Barsanti et al. [16] with minor modifications. A 60 mL aliquot of each algal culture was centrifuged at 9000 rpm for 15 min. The cell pellet was resuspended in 1 mL of distilled water and frozen at −20 °C for 12 h to facilitate cell disruption. The thawed suspension was treated with 1% (*w*/*v*) sodium dodecyl sulfate and 5% (*w*/*v*) Na_2_EDTA at 37 °C for 1 h to solubilize the cellular components other than β-1,3-glucan. The mixture was centrifuged again, and the insoluble pellet containing the β-1,3-glucan was washed repeatedly with hot distilled water (70 °C) until the supernatant became clear. The final pellet was dried overnight at 60 °C, and the β-1,3-glucan yield was gravimetrically determined. The intracellular content of β-1,3-glucan was quantified by dividing it by the number of cells.

### 2.10. Statistical Analysis

All experiments were performed in triplicate, and data are presented as the mean ± standard deviation. Statistical significance was determined using one-way ANOVA followed by Tukey’s HSD test, with *p* < 0.05 being considered significant. Statistical analyses were performed using SPSS software (version 25.0, IBM Corp., Armonk, NY, USA), and graphs were generated using Origin software (version 2021, OriginLab Corp., Northampton, MA, USA).

## 3. Results and Discussion

### 3.1. Effect of Different Treatment Methods on the E. gracilis Growth in Three Fruit Waste Media

The growth of *E. gracilis* was evaluated in three different fruit waste media (SB, BP, and WP) under four treatment conditions (control, WE, HTP, and DSA). The growth curves, μ, and CDT were analyzed to assess the effectiveness of each treatment method (Figure 1).

The DSA treatment consistently outperformed the other treatments across all media. For SB, the DSA treatment resulted in the highest cell density by Day 14, reaching 2.08 times that of the control group. This treatment also yielded the highest specific growth rate (0.22 d^−1^) and shortest cell doubling time (3.15 d). The WE and HTP treatments showed no significant improvement over the control. Similar trends were observed in the BP and WP treatments, albeit with varying degrees of improvement. In a BP medium, the DSA treatment led to a cell density that was 1.41 and 1.36 times higher than the WE and HTP treatments, respectively, after 14 d with the most favorable growth rate (0.16 d^−1^) and doubling time (4.33 d). For the WP medium, despite a slow initial growth, the DSA treatment group achieved the highest cell density by Day 14 of 1.70 times that of the control group. The specific growth rate (0.17 d^−1^) and cell doubling time (3.98 d) under DSA treatment were superior to the other treatments and control group.

The superior performance of the DSA treatment across all three fruit waste media can be attributed to its effective breakdown of complex carbohydrates, releasing readily available nutrients, particularly simple sugars, which *E. gracilis* can easily assimilate for growth [17]. The hydrolysis of cellulose and hemicellulose in the fruit waste by DSA treatment has been reported to increase the availability of fermentable sugars [7]. The varying degrees of improvement observed among the three fruit waste media suggest differences in their composition and the effectiveness of the DSA treatment on each substrate. SB, being rich in cellulose and hemicellulose [5], appeared to be most responsive to the DSA treatment, possibly due to the more efficient liberation of fermentable sugars. HTP’s moderate performance indicates that high temperature and pressure alone are less effective than acid hydrolysis in breaking down complex carbohydrates; though it is still an improvement over the control and WE treatments [6]. WE’s minimal improvement suggests water-soluble nutrients alone are insufficient for significant *E. gracilis* growth enhancement.

While our results demonstrate the potential of fruit waste as a growth medium for *E. gracilis*, it is crucial to critically examine the overall cost-effectiveness and sustainability of this approach. The DSA method shows promise by significantly improving biomass and high-value compound yields, potentially enhancing overall energy efficiency. However, the process of preparing fruit waste media involves multiple steps, each incurring costs in terms of energy, water, chemicals, and labor. These costs could potentially be offset by the low or negative cost of fruit waste as a raw material, reduced waste disposal costs for fruit processors, and the production of high-value compounds. Moreover, the valorization of fruit waste aligns with circular economy principles by reducing the demand for virgin resources. Future research should focus on optimizing the preparation process to improve both economic viability and sustainability. This could include exploring less energy-intensive methods, investigating the use of wet biomass, minimizing acid consumption, and considering more environmentally friendly pretreatment alternatives. A comprehensive techno-economic analysis would be necessary to accurately determine the scalability and long-term feasibility of this approach [8].

### 3.2. Analysis of the Reducing Sugar Content in Culture Media

The reducing sugar content in the culture media is a crucial factor influencing the growth of *E. gracilis* as it serves as a primary carbon source. Table 1 illustrates the changes in reducing the sugar content over the 14-day cultivation period under different treatments. In all of the three fruit waste media, the DSA treatment initially resulted in the highest reducing sugar content, followed by HTP and then WE. This trend aligns with the growth performance observed in Section 3.1. The effectiveness of DSA varied among the three fruit wastes due to their distinct chemical compositions. SB contains approximately 40–50% cellulose, 25–35% hemicellulose, and 15–20% lignin [5]. This high lignocellulosic content explains why DSA was particularly effective on SB, resulting in the highest initial reducing sugar content among the three wastes. However, BP has a different composition. The lower lignocellulosic content of BP resulted in a moderate initial reducing sugar content after DSA treatment, but the presence of free sugars contributed to sustained *E. gracilis* growth. WR contains significant amounts of pectin, and its efficiency was relatively low under acidic hydrolysis, resulting in the lowest reducing sugar content after DSA treatment.

The remaining reducing sugar content of SB-DSA remained notably high upon the end of cultivation, yet the cells had entered a stationary phase, primarily due to the limitations in other trace nutrients. Interestingly, the rate of reducing sugar depletion in the SB-DSA was slower compared to the BP-DSA, despite showing the highest growth rates. This phenomenon can be attributed to the high sucrose content that is naturally present in SB [18]. The presence of sucrose provides an additional carbon source for *E. gracilis* growth, resulting in a slower apparent depletion of reducing sugars in the SB treatments. The difference in the sugar composition among the three fruit waste media highlighted the impact of the substrate characteristics on *E. gracilis* growth and sugar utilization patterns. The DSA treatment was generally the most effective in liberating reducing sugars, but DSA pretreatment can also lead to the formation of compounds such as furfural and hydroxymethylfurfural, which have inhibitory effects on microbial growth. In this study, the DSA treatment appeared to strike an optimal balance between sugar liberation and potential inhibitor formation, resulting in the most favorable growth conditions among the treatments studied.

### 3.3. Effect of the Fruit Waste Extracts on E. gracilis Cell Morphology

Cell morphology is a crucial indicator of *E. gracilis* growth and metabolic status [4]. Our analysis revealed significant changes in the cell morphology across the different fruit waste extracts and treatment methods (Figure 2). In the SB extract, the DSA treatment resulted in cells with an AR 1.48 times higher than the control group. The cell PA in SB-DSA was 1.30 times larger than the control. Similar trends were observed in the BR and WR extracts. These morphological changes, particularly the increased cell size and elongation, are indicative of enhanced metabolic activity and a more active physiological state [4]. Interestingly, the cells in all DSA-treated extracts showed a trend toward an increased AR compared to the control and other treatments. This elongation might be a response to the altered nutrient composition in DSA-treated media, possibly reflecting changes in the cell’s swimming behavior or an adaptation to an optimized nutrient uptake [19]. The HTP and WE treatments also induced morphological changes, albeit to a lesser extent than the DSA treatment. This gradient of effect aligns with our observations on growth rates and biomass production, suggesting a strong correlation between cell morphology and overall cellular performance.

These morphological adaptations underscore the profound impact of fruit waste extracts, particularly when subjected to DSA treatment, on *E. gracilis* physiology. The observed changes not only indicate enhanced growth, but also suggest potential alterations in cellular metabolism and storage product accumulation, which may have significant implications for the biotechnological applications of *E. gracilis* cultivated in these media [3].

### 3.4. Effect of the Fruit Waste Extracts on Photosynthetic Pigment Production in E. gracilis

The impact of various fruit waste extracts and treatment methods on the production of photosynthetic pigments in *E. gracilis* was investigated (Figure 3a,c,e). Chlorophyll a, chlorophyll b, and carotenoids play crucial roles in photosynthesis, light absorption, and photoprotection [20]. The production of these pigments varied significantly across different fruit waste substrates and treatment methods, with a general trend of increased production under more intensive treatments.

For all three substrates, the DSA treatment consistently yielded the highest pigment concentrations, which aligns with the trend observed in cell biomass production. In the SB medium, chlorophyll a increased from 0.81 mg·L^−1^ (WE) to 2.81 mg·L^−1^ (DSA), chlorophyll b from 0.23 mg·L^−1^ (WE) to 0.62 mg·L^−1^ (DSA), and carotenoids from 0.41 mg·L^−1^ (WE) to 0.89 mg·L^−1^ (DSA). Notably, SB-DSA produced the highest chlorophyll a content among all of the treatments, suggesting its particular effectiveness in enhancing the production of this crucial photosynthetic pigment. In the BP medium, DSA treatment resulted in the highest production of all pigments, with chlorophyll a showing a 1.69-fold increase compared to the control. Chlorophyll b increased significantly from 0.26 mg·L^−1^ (control) to 0.59 mg·L^−1^ (DSA), while carotenoid production was 1.56-fold higher than the control under DSA treatment. For the WP substrate, all of the treatments significantly increased pigment production compared to the control. DSA treatment showed the highest increases: chlorophyll a from 1.39 mg·L^−1^ (control) to 2.36 mg·L^−1^, chlorophyll b from 0.25 mg·L^−1^ to 0.57 mg·L^−1^ (2.28-fold increase), and carotenoids from 0.43 mg·L^−1^ to 0.96 mg·L^−1^. Interestingly, the HTP treatment also showed improved pigment yields compared to the WE treatment across all substrates, indicating that both heat and acid treatments can enhance the availability of nutrients that support pigment production.

These findings demonstrate that DSA-treated fruit waste extracts significantly enhance the photosynthetic pigment production in *E. gracilis* compared to other treatments and conventional media. This approach could potentially reduce cultivation costs while improving pigment yields, which is valuable for various applications such as biofuel production or high-value pigment synthesis [2].

### 3.5. The Cellular Content and Ratios of Photosynthetic Pigments in E. gracilis

The pigment content per cell (Figure 3b,d,f) provided insights into the physiological state of the *E. gracilis* under different treatments. In the SB medium, the HTP treatment resulted in the highest cellular content of all three pigments, with increases of 1.13-fold, 1.27-fold, and 1.26-fold for the chlorophyll a, chlorophyll b, and carotenoids, respectively, compared to the control. This suggests enhanced light energy absorption and utilization efficiency under HTP treatment [13]. In the BP medium, the HTP treatment also yielded the highest cellular content of chlorophyll a and carotenoids (1.34-fold and 1.24-fold increase, respectively), while the DSA treatment resulted in the highest chlorophyll b content (1.69-fold increase). For the WP substrate, the HTP treatment led to the highest cellular content of all three pigments, with increases of 1.52-fold, 1.42-fold, and 2.02-fold for chlorophyll a, chlorophyll b, and carotenoids, respectively.

The chlorophyll a/b ratio and total chlorophyll/carotenoid ratio provided valuable information about the light-harvesting capacity, dark adaptation ability, and photosynthetic intensity of algal cells [13,21]. These ratios varied across different fruit waste substrates and treatments (Figure 4). In the SB medium, the chlorophyll a/b ratio was highest in the HTP treatment group among the three treatments, although it was lower than the control. The total chlorophyll/carotenoid ratio increased with treatment intensity, reaching its peak under the DSA treatment (3.84 compared to 2.56 in the WE treatment). For the BP substrate, the HTP treatment resulted in the highest chlorophyll a/b ratio (1.10-fold increase compared to control), suggesting a potential optimization of the chlorophyll composition for improved photosynthetic efficiency.

The total chlorophyll/carotenoid ratio was highest in the DSA treatment, indicating the strongest photosynthetic rate under this condition. In the WP medium, the HTP treatment yielded the highest chlorophyll a/b ratio among the treatments. The WE treatment resulted in the highest total chlorophyll/carotenoid ratio among the treatments, although all of the treatment groups showed lower ratios compared to the control.

These variations in the pigment ratios suggest adaptive responses to changing nutrient and light conditions [22]. The findings demonstrate that different fruit waste substrates and treatment methods significantly influence not only the production, but also the composition and cellular content of the photosynthetic pigments in *E. gracilis*, providing valuable insights for optimizing cultivation strategies.

### 3.6. Chlorophyll Fluorescence Analysis of the E. gracilis Grown in Fruit Waste Extracts

Chlorophyll fluorescence analysis is a powerful, non-invasive technique for assessing the photosynthetic performance and stress responses in photosynthetic organisms [23]. This method provides valuable insights into the efficiency of Photosystem II (PSII) and the overall photosynthetic capacity [24]. In this study, we examined two key parameters: the maximum quantum yield of PSII (Fv/Fm) and the ratio of the variable to initial fluorescence (Fv/F0). These parameters indicate the maximum photochemical efficiency and potential activity of PSII, respectively [25].

As illustrated in Figure 5, the Fv/Fm and Fv/F0 values exhibited significant variation across the different fruit waste substrates and treatments. In the SB medium, the HTP treatment group demonstrated the highest Fv/Fm and Fv/F0 values, indicating superior maximum photochemical efficiency and potential activity. The Fv/Fm value increased from 0.39 in the control group to 0.56 in the HTP group, while the Fv/F0 value rose from 0.65 to 1.24, representing increases of 1.44-fold and 1.91-fold, respectively. The DSA group showed lower Fv/Fm and Fv/F0 values compared to the HTP group, but these values were still higher than those in the control group. Similar trends were also observed with the BP medium, and the HTP treatment group again demonstrated the highest Fv/Fm and Fv/F0 values. Interestingly, in the WP medium, the DSA treatment group exhibited the highest Fv/Fm and Fv/F0 values, which were 1.20 times and 1.98 times higher than those of the HTP group, respectively.

These results demonstrate that the photosynthetic performance of *E. gracilis* varies considerably depending on the fruit waste substrate and treatment method. The enhanced Fv/Fm and Fv/F0 values observed in certain treatments suggest improved photosynthetic efficiency, which may correlate with the changes in pigment production noted in earlier sections. This finding aligns with previous studies that have reported a strong relationship between chlorophyll fluorescence parameters and photosynthetic pigment content in microalgae [26]. The observed variations in photosynthetic efficiency across different treatments could be attributed to changes in the availability of nutrients or the presence of inhibitory compounds in the fruit waste media [27]. Furthermore, the differential responses to the HTP and DSA treatments among the fruit waste types suggest that the optimal pretreatment method may be substrate-specific, highlighting the importance of tailored approaches in biomass utilization.

### 3.7. Effects of the Fruit Waste Treatments on the Paramylon Production in E. gracilis

β-1,3-glucan, also known as paramylon, is a unique storage polysaccharide found in *Euglena* species, and it has various potential applications in the food and pharmaceutical industries [28]. This study investigated the effects of different fruit waste treatments on both the total and cellular β-1,3-glucan content in *E. gracilis* (Figure 6).

Across all of the fruit waste media, the DSA treatment consistently resulted in the highest total β-1,3-glucan content. These increases were statistically significant compared to the control, with fold changes of 1.25, 1.13, and 1.25 for the SB, BP, and WP media, respectively. This consistent trend suggests that DSA treatment may enhance the overall β-1,3-glucan production in *E. gracilis*, regardless of the specific fruit waste substrate. Interestingly, while the total β-1,3-glucan content increased, the cellular β-1,3-glucan content showed a contrasting trend. In all of the fruit waste media, the DSA treatment led to a significant decrease in the cellular β-1,3-glucan content compared to the control. The cellular β-1,3-glucan content decreased by 39.08%, 17.52%, and 27.49% for the SB, BP, and WP media, respectively, compared to the control. This inverse relationship between the total and cellular β-1,3-glucan content suggests that, under DSA treatment, *E. gracilis* may allocate more energy toward cell proliferation rather than individual cell metabolite accumulation [29]. The WE and HTP treatments showed less pronounced effects on the β-1,3-glucan content. In most cases, these treatments did not result in significant differences in the total β-1,3-glucan content compared to the control. However, in the WR medium, all of the treatments led to significantly lower cellular β-1,3-glucan content than the control, indicating that the effect of these treatments may be substrate-dependent.

These results demonstrate that fruit waste treatments, particularly DSA, can significantly influence the β-1,3-glucan production in *E. gracilis*. The observed increase in the total β-1,3-glucan content coupled with the decreased cellular content in most treatments suggests that these conditions may promote both cell proliferation and the overall β-1,3-glucan production [30]. This finding could have important implications for optimizing *E. gracilis* cultivation for paramylon production.

## 4. Conclusions

This study demonstrates the effective valorization of fruit waste for *E. gracilis* cultivation and high-value compound production. DSA hydrolysis consistently outperformed the other pretreatment methods, significantly enhancing *E. gracilis* growth, photosynthetic pigment production, and β-1,3-glucan accumulation across all fruit waste media. These findings highlight the potential of pretreated fruit waste as a cost-effective and sustainable medium for *E. gracilis* cultivation. However, scaling these processes presents challenges, including economic feasibility, equipment requirements, and potential variability in fruit waste composition. Despite these hurdles, the improved yields of the biomass and valuable compounds could offset costs and provide a competitive advantage in sustainability. The bioproducts obtained have potential applications in food and cosmetic industries, aligning with circular economy principles. Future research should focus on optimizing pretreatment processes, conducting pilot-scale studies, and exploring the bioactivity and safety of produced compounds. By addressing these aspects, fruit waste valorization through *E. gracilis* cultivation could significantly contribute to sustainable biorefinery processes and the circular bioeconomy.

## Figures and Tables

**Figure 1 foods-13-03439-f001:**
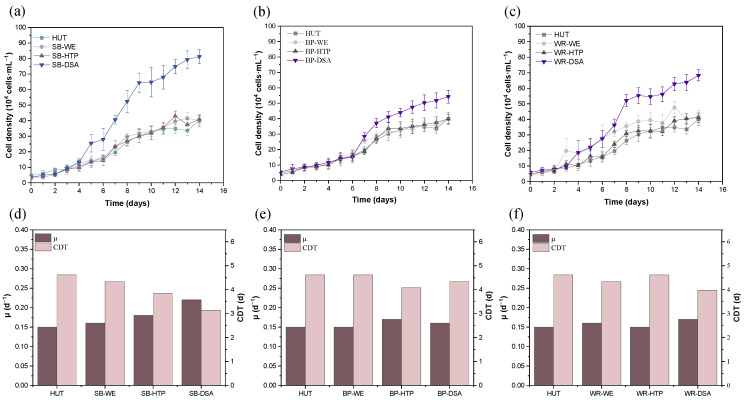
Growth performance of the *E. gracilis* in different fruit waste media under various treatments. (**a**–**c**) the growth curves in SB, BP, and WR, respectively; and (**d**–**f**) the μ and CDT in SB, BP, and WR, respectively. The error bars represent the standard deviation (n = 3).

**Figure 2 foods-13-03439-f002:**
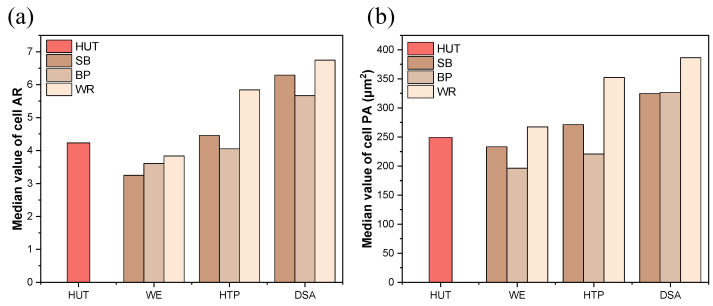
The cell morphology of *E. gracilis* in the different fruit waste media under various treatments. (**a**) Median value of cell AR. (**b**) Median value of cell PA. The values are represented by the median (n > 100).

**Figure 3 foods-13-03439-f003:**
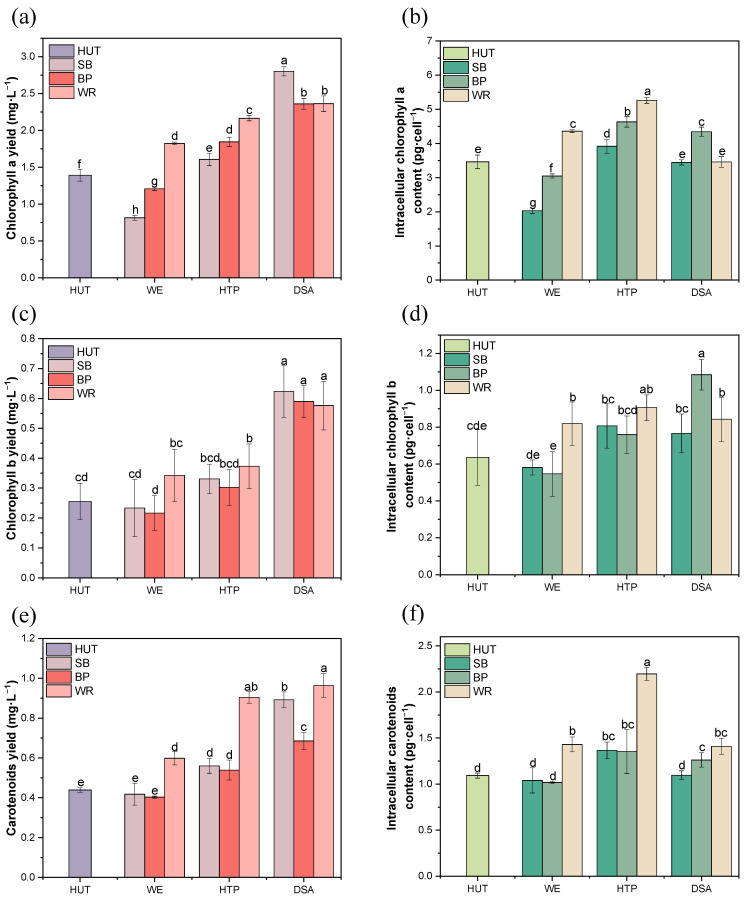
The photosynthetic pigment production and cellular content in the *E. gracilis* cultivated in the different fruit waste media under various treatments. (**a**,**c**,**e**) The yield of chlorophyll a, chlorophyll b, and carotenoids, respectively; (**b**,**d**,**f**) the cellular content of chlorophyll a, chlorophyll b, and carotenoids, respectively. Error bars represent the standard deviation (n = 3). Different lowercase letters indicate significant differences among groups (*p* < 0.05).

**Figure 4 foods-13-03439-f004:**
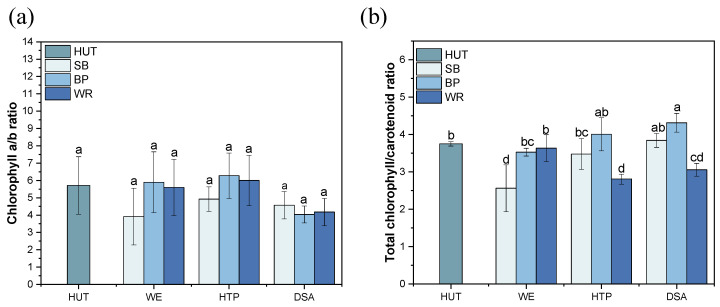
The chlorophyll a/b ratio and total chlorophyll/carotenoid ratio in the *E. gracilis* cultivated in the different fruit waste media under various treatments. (**a**) the chlorophyll a/b ratio, and (**b**) the total chlorophyll/carotenoid ratio. Error bars represent the standard deviation (n = 3). Different lowercase letters indicate significant differences among groups (*p* < 0.05).

**Figure 5 foods-13-03439-f005:**
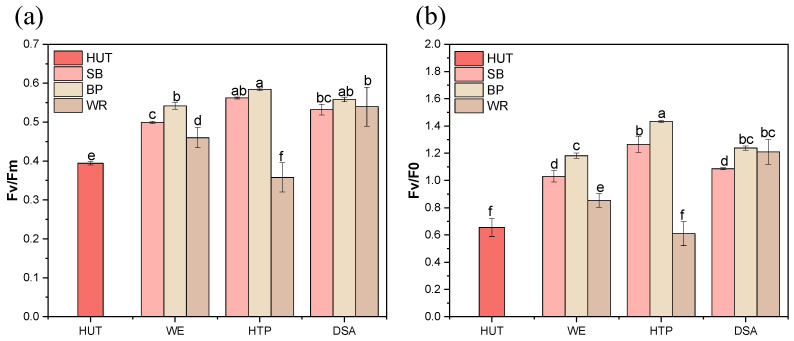
The photosynthetic efficiency parameters of the *E. gracilis* cultivated in the different fruit waste media under various treatments. (**a**) The maximum quantum yield of PSII (Fv/Fm), and (**b**) the potential photosynthetic capacity (Fv/F0). Error bars represent the standard deviation (n = 3). Different lowercase letters indicate significant differences among groups (*p* < 0.05).

**Figure 6 foods-13-03439-f006:**
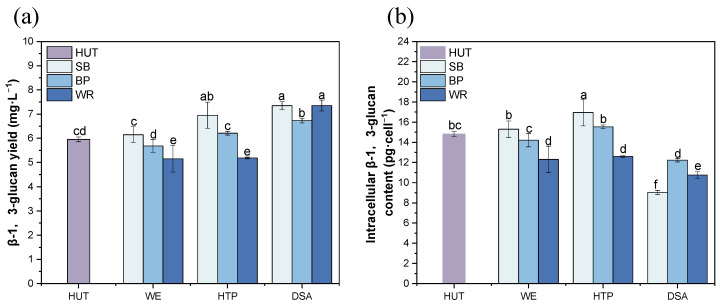
β-1,3-glucan production by the *E. gracilis* cultivated in the different fruit waste media under various treatments. (**a**) β-1,3-glucan yield, and (**b**) intracellular β-1,3-glucan content. Error bars represent the standard deviation (n = 3). Different lowercase letters indicate significant differences among groups (*p* < 0.05).

**Table 1 foods-13-03439-t001:** The reducing sugar content and consumption rate in the different fruit waste media under various treatments during *E. gracilis* cultivation.

Fruit Waste Types	Different Treatments Applied	Reducing Sugar Content at the Start (g/100 mL) Day 0	Reducing Sugar Content at the End (g/100 mL) Day 14	Rate of Consumption of Reducing Sugar (mg·d^−1^)
SB	WE	0.26 ± 0.03 ^e^	0.21 ± 0.02 ^d^	0.0036 ± 0.0005 ^f^
	HTP	0.51 ± 0.05 ^d^	0.22 ± 0.03 ^d^	0.0207 ± 0.0020 ^e^
	DSA	4.23 ± 0.21 ^a^	2.89 ± 0.15 ^a^	0.0957 ± 0.0048 ^b^
BP	WE	1.20 ± 0.10 ^c^	0.20 ± 0.02 ^d^	0.0714 ± 0.0036 ^c^
	HTP	1.43 ± 0.12 ^c^	0.38 ± 0.04 ^c^	0.0750 ± 0.0038 ^c^
	DSA	3.05 ± 0.15 ^b^	0.48 ± 0.05 ^b^	0.1835 ± 0.0092 ^a^
WR	WE	1.70 ± 0.14 ^c^	0.52 ± 0.05 ^b^	0.0840 ± 0.0042 ^bc^
	HTP	1.76 ± 0.15 ^c^	0.57 ± 0.06 ^b^	0.0850 ± 0.0043 ^bc^
	DSA	1.85 ± 0.16 ^c^	0.61 ± 0.06 ^b^	0.0880 ± 0.0044 ^bc^

Note: Values are presented as the mean ± standard deviation (n = 3). Different superscript letters within the same column indicate statistically significant differences (*p* < 0.05).

## Data Availability

The original contributions presented in the study are included in the article/Appendix A, further inquiries can be directed to the corresponding author.

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
