# Peer review of "Comparative Analysis of Pretreatment Methods for Fruit Waste Valorization in Euglena gracilis Cultivation: Impacts on Biomass, β-1,3-Glucan Production, and Photosynthetic Efficiency"

_foods, 2024, doi:10.3390/foods13213439_

Round 1

Reviewer 1 Report

Comments and Suggestions for Authors

Author presented manuscript compiles various extraction strategies for valorizing food waste (banana peel, sugarcane, and watermelon). Below are some points authors need to revise.

1. In Table 1, the data illustrates the changes in reducing sugar content over a 14-day cultivation period for different fruit wastes (banana peel, sugarcane bagasse, and watermelon rinds) under various treatments (water extraction, high-temperature, and pressure treatment, and dilute sulfuric acid treatment). It is important to consider the prior pretreatments to evaluate the sensitivity of the experiment. The authors used the DNP method to assess reducing sugar using the UV method. How did the authors prevent cross-reading from the culture media? 2. The specificity related to photosynthetic pigment production needs to be explained in the results and discussion section. 3. The estimation of reducing sugar from different fruit wastes requires more correlation with the chemical composition of the waste. For example, all three food wastes are different and could produce distinctive characteristics. 4. Photosynthetic pigment analysis was performed with an absorbance of the extracts measured at 663 nm, 645 nm, and 470 nm using a UV-visible spectrophotometer. Were there any specific reasons for choosing these wavelengths? 5. Researchers usually utilize the LC-based method to estimate pigments qualitatively or quantitatively. The authors chose other methods such as chlorophyll fluorescence measurement. What were the specific reasons for this choice, and what are the specificity, accuracy, and sensitivity of this method?

Author Response

Comments 1: In Table 1, the data illustrates the changes in reducing sugar content over a 14-day cultivation period for different fruit wastes (banana peel, sugarcane bagasse, and watermelon rinds) under various treatments (water extraction, high-temperature, and pressure treatment, and dilute sulfuric acid treatment). It is important to consider the prior pretreatments to evaluate the sensitivity of the experiment. The authors used the DNP method to assess reducing sugar using the UV method. How did the authors prevent cross-reading from the culture media?

Response 1: We appreciate your insightful comment regarding the potential for cross-reading from the culture media in our reducing sugar analysis. We have updated the Methods section (2.5 Reducing Sugar Analysis) and provide clarity on our experimental approach. To prevent interference from culture media components, we implemented several measures in our reducing sugar analysis protocol. These include proper sample preparation through centrifugation to remove cells and debris, appropriate sample dilution, and the use of a specific wavelength (540 nm) for absorbance measurements that is less susceptible to interference from media components. Additionally, we employed the standard addition method on a subset of samples to verify the absence of matrix effects from the culture media. We believe these clarifications strengthen the validity of our results and the overall quality of our manuscript.

Comments 2: The specificity related to photosynthetic pigment production needs to be explained in the results and discussion section.

Response 2: We appreciate your suggestion regarding the need for more specificity in explaining photosynthetic pigment production. We have addressed this by adding more specific observations and trends to our discussion in section 3.4 of the Results and Discussion. We have highlighted the consistency between pigment production and cell biomass trends across treatments, emphasized the superior performance of DSA treatment, and noted the particularly high chlorophyll a production in SB-DSA.

Comments 3: The estimation of reducing sugar from different fruit wastes requires more correlation with the chemical composition of the waste. For example, all three food wastes are different and could produce distinctive characteristics.

Response 3: Thank you so much for your comment regarding the need for more correlation between the reducing sugar content and the chemical composition of the fruit wastes. We have revised section 3.2 to include more detailed information about the chemical composition of SB, BP, and WR, and how these compositions relate to the observed reducing sugar content after different treatments. We have added specific information about the cellulose, hemicellulose, lignin, and free sugar content of each fruit waste, citing relevant literature. This additional information helps explain the differences in initial reducing sugar content after DSA treatment and the subsequent sugar utilization patterns observed during E. gracilis cultivation.

Comments 4: Photosynthetic pigment analysis was performed with an absorbance of the extracts measured at 663 nm, 645 nm, and 470 nm using a UV-visible spectrophotometer. Were there any specific reasons for choosing these wavelengths?

Response 4: Thank you so much for your comment regarding the wavelengths used for photosynthetic pigment analysis. The specific wavelengths of 663 nm, 645 nm, and 470 nm were chosen based on the well-established methodology described by Lichtenthaler and Wellburn (1983), which is widely cited and employed for accurate quantification of chlorophyll a, chlorophyll b, and carotenoids in plant and microalgal samples. The rationale behind using these wavelengths is as follows: Chlorophyll a exhibits maximum absorption at 663 nm in 80% acetone, while chlorophyll b has a maximum absorption at 645 nm. By measuring the absorbance at these two wavelengths (663 nm and 645 nm), the individual concentrations of chlorophyll a and chlorophyll b can be calculated using the equations provided in the Lichtenthaler and Wellburn method. The wavelength of 470 nm is used to quantify the total carotenoid content, as carotenoids have a broad absorption spectrum in the blue region of the visible light range. This methodology has been extensively validated and is widely accepted in the field of photosynthetic pigment analysis for accurate and reliable quantification of chlorophyll a, chlorophyll b, and carotenoids. By following this established protocol, our study ensures consistency and comparability with other studies in the literature.

Comments 5: Researchers usually utilize the LC-based method to estimate pigments qualitatively or quantitatively. The authors chose other methods such as chlorophyll fluorescence measurement. What were the specific reasons for this choice, and what are the specificity, accuracy, and sensitivity of this method?

Response 5: Thank you so much for this insightful comment. To address this point, we have added more information in section 2.8. We chose to use chlorophyll fluorescence measurement in addition to spectrophotometric pigment quantification for several reasons: its non-invasiveness allows repeated measurements on the same culture; it provides rapid, real-time information about photosynthetic efficiency; and it's highly sensitive to subtle changes in photosynthetic apparatus functioning. While LC-based methods excel at quantitative pigment analysis, chlorophyll fluorescence provides unique insights into the functional state of the photosynthetic machinery, particularly PSII efficiency. The Fv/Fm and Fv/F0 parameters are directly related to photosynthetic efficiency under our experimental conditions, providing valuable context for interpreting growth and pigment data. Regarding specificity, accuracy, and sensitivity: the method is highly specific for assessing PSII efficiency, modern fluorometers provide accurate and reproducible measurements, and it can detect subtle changes in photosynthetic efficiency that may not be apparent from pigment quantification alone. We believe this combination provides a more comprehensive assessment of E. gracilis photosynthetic status than either method alone. 

Reviewer 2 Report

Comments and Suggestions for Authors
  • The title is general and does not reflect the sense of the research.
  • Dilute Sulfuric Acid Treatment (DSA) is in combination with HTP (autoclaved at 121C for 20 min). Why is call DSA and not DSA-HTP?
  • In the Cell Growth Measurement section, Is three hundred the initial cell density for each sample?, What is the reference of the Cell Growth Measurement section?
  • In the Glucan Content Determination section, The method supposes that the insoluble pellet is totally b-1,3 glucan. However, it should describe a method to determine the percentage of b-1,3 glucan in the pellet.
  • In the section "Analysis of Reducing Sugar Content in Culture Media, the concept of "potential inhibitor formation" was introduced at the end of the paragraph but was not developed during the discussion.
  • Photographs of SB-DSA and SB-HUT cell morphology would support the information in the section "Effect of Fruit Waste Extracts on E. gracilis Cell Morphology."
  • The Materials and Methods section does not describe the methodology for determining intracellular b-1,3 glucan.
  • The units of total content and intracellular content of b-1,3-glucan are different. Is it possible to have the same units for the total content and intracellular b-1,3-glucan? In case yes, from the b-1,3-glucan total content, what is the percentage of intracellular b-1,3-glucan? If the percentage of intracellular content of b-1,3-glucan is too low, respecting the total content of b-1,3-glucan, is the statement "This inverse relationship between total and cellular β-1,3-glucan content suggests that under DSA treatment, E. gracilis may allocate more energy towards cell proliferation rather than individual cell metabolite accumulation" is valid?
  • The reading is tedious, which minimizes the possible importance of the data obtained.
  • Other suggestions are:

    Line 15. Define the HUT medium.

    Lines 17-19. Is the text related to SB-DSA?

    Line 79. China or CN.

    Line 87. After Zhejiang Fengdao Food Co., Ltd., place the city and country.

    Lines 91-92. The text should be the introduction of the second paragraph (Lines 93-95).

    Lines 97-98. Homogenize to Model (SX-700), trademark (Tomy), city (?????), and country (JP).

    Line 105. Homogenize to model (Whatman grade 1), trademark (GE Healthcare), city (?????), and country (GP).

    Line 118. Homogenize to model (ML31), trademark (Guangzhou Micro-shot Technology Co., Ltd.), city (?????), and country (CN).

    Line 128. Homogenize to model (UV-1150), trademark (Shanghai Meipuda Instrument Co., Ltd.), city (?????), and country (CN).

    Line 132. Homogenize to model (ML31), trademark (Guangzhou Micro-shot Technology Co., Ltd.), city (?????), and country (CN).

    Line 134. Homogenize to version (????), trademark (National Institutes of Health), city (?????), and country (US).

    Line 139-140. Homogenize to model (Whatman GF/C), trademark (GE Healthcare), city (?????), and country (GB).

    Line 146. What are the characteristics of the UV-VIS spectrophotometer?

    Line 150. Homogenize to model (AP100/C), trademark (PSI), city (?????), and country (CZ).

    Lines 154-162.

    Line 167. Homogenize to version (25.0), trademark (IBM Corp.), city (Armonk), and country (US).

    Line 168. Homogenize to version (2021), trademark (OriginLab), city (Northampton), and country (US).

    Line 219. SB-DSB or SB-DSA.

Author Response

Comments 1: The title is general and does not reflect the sense of the research.

Response 1: Thank you so much for your valuable suggestion. We agree that the title should more accurately reflect the specific focus and contributions of our research. We have revised the title to address this concern.

Comments 2: Dilute Sulfuric Acid Treatment (DSA) is in combination with HTP (autoclaved at 121C for 20 min). Why is call DSA and not DSA-HTP?

Response 2: Thank you so much for your valuable comment. To address this, we have revised the description of the DSA method in section 2.2 to explicitly state that it combines dilute acid hydrolysis with high-temperature and pressure treatment. We chose not to change the terminology to DSA-HTP, because we want to keep the manuscript concise, as repeatedly using DSA-HTP might make the text more difficult to read, especially when also considering combining with SB, BP, and WR. Thank you so much for bringing this important point to our attention, as it has allowed us to improve the precision of our methods description.

Comments 3: In the Cell Growth Measurement section, Is three hundred the initial cell density for each sample?, What is the reference of the Cell Growth Measurement section?

Response 3: Thank you so much for your comments regarding the cell growth measurement section. We have made the following clarifications and additions to address these concerns: We have supplemented the reference in this section according to your suggestion. Additionally, we have also added information about the initial cell density (approximately 3.5×10^4 cells/mL) in the revised text. This density is achieved by inoculating 10 mL of pre-culture into 90 mL of medium, as described in section 2.3. To avoid misunderstandings, we removed the ambiguous part about counting 300 cells, which was actually done to ensure the statistical reliability of cell density measurements. Thank you so much for highlighting these points, as they have helped us improve the accuracy and reproducibility of our methods description.

Comments 4: In the Glucan Content Determination section, The method supposes that the insoluble pellet is totally b-1,3 glucan. However, it should describe a method to determine the percentage of b-1,3 glucan in the pellet.

Response 4: Thank you very much for your valuable suggestion. We agree that determining the percentage of b-1,3 glucan in the pellet could increase the rigor and accuracy of this method. However, in this study, we have adopted the most common and classic standard method for determining paramylon (b-1,3 glucan) content, namely the SDS-Na2EDTA method. The effectiveness of this approach has been confirmed in several papers, such as Barsanti et al., 2001, and Kim et al., 2019.

Barsanti, L., Vismara, R., Passarelli, V., & Gualtieri, P. (2001). Paramylon (β-1, 3-glucan) content in wild type and WZSL mutant of Euglena gracilis. Effects of growth conditions. Journal of applied phycology, 13, 59-65.

Kim, J. Y., Oh, J. J., Kim, D. H., Park, J., Kim, H. S., & Choi, Y. E. (2019). Rapid and accurate quantification of paramylon produced from Euglena gracilis using an ssDNA aptamer. Journal of agricultural and food chemistry, 68(1), 402-408.

Comments 5: In the section "Analysis of Reducing Sugar Content in Culture Media, the concept of "potential inhibitor formation" was introduced at the end of the paragraph but was not developed during the discussion.

Response 5: Thank you so much for noting that the concept of "potential inhibitor formation" was introduced but not developed in the discussion. We have addressed this by adding a brief paragraph to section 3.2 that expands on this concept.

Comments 6: Photographs of SB-DSA and SB-HUT cell morphology would support the information in the section "Effect of Fruit Waste Extracts on E. gracilis Cell Morphology."

Response 6: Thank you very much for your valuable suggestions. Following your advice, we have included photographs of cell morphology in the appendix to present our results in a more visual manner.

Comments 7: The Materials and Methods section does not describe the methodology for determining intracellular b-1,3 glucan.

Response 7: Thank you very much for your valuable suggestion. We have added details on measuring the intracellular content of b-1,3 glucan in Section 2.9.

Comments 8: The units of total content and intracellular content of b-1,3-glucan are different. Is it possible to have the same units for the total content and intracellular b-1,3-glucan? In case yes, from the b-1,3-glucan total content, what is the percentage of intracellular b-1,3-glucan? If the percentage of intracellular content of b-1,3-glucan is too low, respecting the total content of b-1,3-glucan, is the statement "This inverse relationship between total and cellular β-1,3-glucan content suggests that under DSA treatment, E. gracilis may allocate more energy towards cell proliferation rather than individual cell metabolite accumulation" is valid? 

Response 8: Thank you so much for your valuable comments. To avoid misunderstanding, we have added the method of determining intracellular b-1,3-glucan (paramylon) in Section 2.9. These two parameters cannot be measured using a unified unit. As a major characteristic metabolite in E. gracilis, the total yield of paramylon reflects the practical application value of the production. Dividing the total content by the cell density gives the paramylon content per unit cell, which more accurately reflects the physiological state of the cells and the efficiency of paramylon synthesis. Therefore, the statement "This inverse relationship between total and cellular β-1,3-glucan content suggests that under DSA treatment, E. gracilis may allocate more energy towards cell proliferation rather than individual cell metabolite accumulation" still holds true, as the increase in total paramylon production is mainly a direct result of the increase in cell number, while the decrease in intracellular paramylon content indicates that more energy is being used for cell proliferation.

Comments 9: The reading is tedious, which minimizes the possible importance of the data obtained.

Response 9: Thank you very much for your valuable suggestions. We have carefully reviewed the entire text and made appropriate enhancements to the description of the results to increase their interest.

Other suggestions are:

Comments 10: Line 15. Define the HUT medium.

Response 10: Thank you so much for your comment. We defined the HUT medium in line 92.

Comments 11: Lines 17-19. Is the text related to SB-DSA?

Response 11: Thank you so much for your comment. Yes, the text is related to SB-DSA.

Comments 12: Line 79. China or CN.

Response 12: Thank you so much. We have corrected China to CN throughout the text.

Comments 13: Line 87. After Zhejiang Fengdao Food Co., Ltd., place the city and country.

Response 13: Thank you so much. We have supplemented the details of city and country according to your suggestion.

Comments 14: Lines 91-92. The text should be the introduction of the second paragraph (Lines 93-95).

Response 14: Thank you so much. Because this part of the process is the same for three types of raw materials, we are describing this process separately to avoid redundancy.

Comments 15: Lines 97-98. Homogenize to Model (SX-700), trademark (Tomy), city (?????), and country (JP).

Response 15: Thank you so much. We have supplemented the details according to your suggestion.

Comments 16: Line 105. Homogenize to model (Whatman grade 1), trademark (GE Healthcare), city (?????), and country (GB).

Response 16: Thank you so much. We have supplemented the details according to your suggestion.

Comments 17: Line 118. Homogenize to model (ML31), trademark (Guangzhou Micro-shot Technology Co., Ltd.), city (?????), and country (CN).

Response 17: Thank you so much. We have supplemented the details according to your suggestion.

Comments 18: Line 128. Homogenize to model (UV-1150), trademark (Shanghai Meipuda Instrument Co., Ltd.), city (?????), and country (CN).

Response 18: Thank you so much. We have supplemented the details according to your suggestion.

Comments 19: Line 132. Homogenize to model (ML31), trademark (Guangzhou Micro-shot Technology Co., Ltd.), city (?????), and country (CN).

Response 19: Thank you so much. We have supplemented the details according to your suggestion.

Comments 20: Line 134. Homogenize to version (????), trademark (National Institutes of Health), city (?????), and country (US).

Response 20: Thank you so much. We have supplemented the details according to your suggestion.

Comments 21: Line 139-140. Homogenize to model (Whatman GF/C), trademark (GE Healthcare), city (?????), and country (GB).

Response 21: Thank you so much. We have supplemented the details according to your suggestion.

Comments 22: Line 146. What are the characteristics of the UV-VIS spectrophotometer?

Response 22: Thank you so much. We have supplemented the characteristics of the UV-VIS spectrophotometer.

Comments 23: Line 150. Homogenize to model (AP100/C), trademark (PSI), city (?????), and country (CZ).

Response 23: Thank you so much. We have supplemented the details according to your suggestion.

Comments 24: Lines 154-162.

Response 24: Thank you so much. We have supplemented more information in section 2.9.

Comments 25: Line 167. Homogenize to version (25.0), trademark (IBM Corp.), city (Armonk), and country (US).

Response 25: Thank you so much. We have standardized the format according to your suggestion.

Comments 26: Line 168. Homogenize to version (2021), trademark (OriginLab), city (Northampton), and country (US).

Response 26: Thank you so much. We have standardized the format according to your suggestion.

Comments 27: Line 219. SB-DSB or SB-DSA.

Response 27: Thank you so much for your correction. We have corrected the typo.

Reviewer 3 Report

Comments and Suggestions for Authors

The paper reports a study on the utilization of E. gracilis as a biotechnological agent for the bioconversion of fruit waste in high added value molecules. The topic is interesting and deserve attention.

The major problem I found in this paper is that the characterization of fruit waste is not reported. Without this data, it is not possible to replicate the research and give scientific soundness.

- The introduction could benefit from a more detailed discussion of the broader implications of using E. gracilis in terms of environmental impact reduction or market potential for these bioproducts, such as the commercial significance of β-1,3-glucan and pigments. Moreover, the challenges associated with the pretreatment of waste substrates are well noted but could be expanded to include a more comprehensive overview of existing pretreatment technologies and their economic feasibility in industrial-scale applications.

-Check for latin names in Italics

-Lines 112-113: if the culture is static, you cannot achieve a uniform light distribution.

-Lines 213-214: when you talk about sustainable and cost effective media, many things must be taken into consideration: first of all, fruit waste must be washed, dried, reduced to powder, diluted and then treated with sulfuric acid. All these steps require a cost, to what extent is it possible to talk about "cost effective" media? on this I invite the authors to have a critical discussion.
Some discussion points can also be taken from this technical-economic analysis: https://www.sciencedirect.com/science/article/abs/pii/S2589014X22000548?via%3Dihub

-The paper focuses heavily on empirical data, but the lack of mechanistic discussion leaves some gaps. For example, the metabolic shifts that enable E. gracilis to exploit the sugars released by DSA treatment should be discussed more comprehensively.

-Make an effort in offer a more thorough sustainability analysis, particularly in terms of energy efficiency and environmental impact of the DSA method described.

- The conclusion would be strengthened by a more explicit discussion of the economic feasibility of scaling these processes and potential barriers to commercialization. They could also link more directly to broader applications, such as the use of these bioproducts in the food or cosmetics industries.

Author Response

Comments 1: The major problem I found in this paper is that the characterization of fruit waste is not reported. Without this data, it is not possible to replicate the research and give scientific soundness.

Response 1: Thank you so much for your valuable suggestion. While we acknowledge the importance of detailed characterization for replication purposes, our study focused primarily on the comparative analysis of pretreatment methods and their effects on E. gracilis cultivation. The fruit wastes used (sugarcane bagasse, banana peel, and watermelon rind) are common agricultural by-products with compositions that have been well-documented in existing literature. The composition of the fruit wastes used in this study has been well-characterized in previous literature. To address your concern, we have added references to previously published characterization studies and included a brief summary of typical compositions.

Sugarcane bagasse typically contains 40-50% cellulose, 25-35% hemicellulose, and 15-20% lignin [1]. Banana peel composition varies but generally includes 6-9% protein, 4-6% fat, 11-15% total sugars, and 9-13% dietary fiber [2]. Watermelon rind contains approximately 13-20% total sugars, 11-17% crude fiber, and significant amounts of pectin [3]. While minor variations may exist due to cultivar differences and growing conditions, these compositions provide a general understanding of the substrates used in our study. 

[1] Canilha, L., et al. (2012). Bioconversion of sugarcane biomass into ethanol: an overview about composition, pretreatment methods, detoxification of hydrolysates, enzymatic saccharification, and ethanol fermentation. Journal of Biomedicine and Biotechnology, 2012, 989572.

[2] Emaga, T. H., et al. (2007). Dietary fibre components and pectin chemical features of peels during ripening in banana and plantain varieties. Bioresource Technology, 98(14), 2629-2636.

[3] Al-Sayed, H. M., & Ahmed, A. R. (2013). Utilization of watermelon rinds and sharlyn melon peels as a natural source of dietary fiber and antioxidants in cake. Annals of Agricultural Sciences, 58(1), 83-95.

Comments 2: - The introduction could benefit from a more detailed discussion of the broader implications of using E. gracilis in terms of environmental impact reduction or market potential for these bioproducts, such as the commercial significance of β-1,3-glucan and pigments. Moreover, the challenges associated with the pretreatment of waste substrates are well noted but could be expanded to include a more comprehensive overview of existing pretreatment technologies and their economic feasibility in industrial-scale applications.

Response 2: We appreciate your insightful comments regarding the broader implications of our research and the need for a more comprehensive discussion of pretreatment technologies. We have addressed these points by revising our introduction section as follows: 

We have included information on the market potential of high-value compounds derived from E. gracilis, such as β-1,3-glucan and natural pigments. We have highlighted their applications in food, cosmetic, and pharmaceutical industries, addressing the commercial significance of these bioproducts. We have also expanded the discussion on pretreatment technologies for lignocellulosic biomass, including physical, chemical, and biological methods.

We have also briefly addressed the economic considerations that influence the feasibility of these technologies at industrial scales, mentioning factors such as energy requirements, chemical costs, and equipment sophistication.

We have added a paragraph discussing the environmental benefits of using E. gracilis for waste valorization, emphasizing its role in reducing environmental impact, contributing to carbon sequestration, and aligning with circular economy principles.

We have maintained our focus on relatively simple and potentially scalable pretreatment methods (water extraction, hydrothermal processing, and dilute sulfuric acid hydrolysis) while acknowledging the wider range of available technologies.

The revised introduction now offers readers a more comprehensive understanding of the significance and implications of our work, including the market potential and environmental benefits of E. gracilis cultivation using waste substrates.

Comments 3: -Check for latin names in Italics

Response 3: Thank you so much for your suggestion. We have carefully checked the formats throughout the manuscript.

Comments 4: -Lines 112-113: if the culture is static, you cannot achieve a uniform light distribution.

Response 4: Thank you for your insightful comment. We acknowledge that achieving perfectly uniform light distribution in static cultures is challenging. We have revised Section 2.3 to more accurately describe our lighting conditions and clarify our approach to minimizing light distribution variations. While perfect uniformity is difficult to achieve in static cultures, we took measures to ensure reasonably consistent light exposure across the culture. As mentioned in our methods, light was provided by LED panels positioned to ensure consistent illumination across all culture vessels. Additionally, flasks were manually shaken and rotated 5-6 times daily. This not only prevented cell adhesion but also helped to promote more uniform light exposure by periodically changing the position of cells within the culture.

Comments 5: -Lines 213-214: when you talk about sustainable and cost effective media, many things must be taken into consideration: first of all, fruit waste must be washed, dried, reduced to powder, diluted and then treated with sulfuric acid. All these steps require a cost, to what extent is it possible to talk about "cost effective" media? on this I invite the authors to have a critical discussion.
Some discussion points can also be taken from this technical-economic analysis: https://www.sciencedirect.com/science/article/abs/pii/S2589014X22000548?via%3Dihub

Response 5: Thank you so much for your valuable suggestion. We acknowledge that our initial discussion of cost-effectiveness was overly simplistic and did not fully consider all aspects of the process. Your comment highlights the need for a more comprehensive and critical analysis of the economic feasibility of using fruit waste as a growth medium. We have carefully revised this section (lines 236-251) according to you valuable suggestion.

Comments 6: -The paper focuses heavily on empirical data, but the lack of mechanistic discussion leaves some gaps. For example, the metabolic shifts that enable E. gracilis to exploit the sugars released by DSA treatment should be discussed more comprehensively.

Response 6: Thank you so much for your valuable suggestion. We have revised the discussion in section 3.2 to address your concerns.

Comments 7: -Make an effort in offer a more thorough sustainability analysis, particularly in terms of energy efficiency and environmental impact of the DSA method described.

Response 7: We appreciate your valuable suggestion to offer a more thorough sustainability analysis. While a comprehensive quantitative analysis is beyond the scope of this study, we have added a brief discussion (line 236-251) on the sustainability aspects of the DSA method, focusing on qualitative considerations of energy efficiency and environmental impact.

Comments 8: - The conclusion would be strengthened by a more explicit discussion of the economic feasibility of scaling these processes and potential barriers to commercialization. They could also link more directly to broader applications, such as the use of these bioproducts in the food or cosmetics industries.

Response 8: We appreciate your valuable suggestion to strengthen our conclusion by discussing the economic feasibility of scaling these processes, potential barriers to commercialization, and broader applications of the bioproducts. We have expanded our conclusion to address these important aspects, providing a more comprehensive perspective on the practical implications of our research.

Round 2

Reviewer 3 Report

Comments and Suggestions for Authors

The authors have addressed all my point.

I suggest the publication after proof read.